# Stabilin Receptors: Role as Phosphatidylserine Receptors

**DOI:** 10.3390/biom9080387

**Published:** 2019-08-20

**Authors:** Seung-Yoon Park, In-San Kim

**Affiliations:** 1Department of Biochemistry, School of Medicine, Dongguk University, Gyeongju 38066, Korea; 2Biomedical Research Institute, Korea Institute Science and Technology, Seoul 02792, Korea; 3KU-KIST School, Korea University, Seoul 02841, Korea

**Keywords:** stabilin-1, stabilin-2, efferocytosis, phosphatidylserine, fusion

## Abstract

Phosphatidylserine is a membrane phospholipid that is localized to the inner leaflet of the plasma membrane. Phosphatidylserine externalization to the outer leaflet of the plasma membrane is an important signal for various physiological processes, including apoptosis, platelet activation, cell fusion, lymphocyte activation, and regenerative axonal fusion. Stabilin-1 and stabilin-2 are membrane receptors that recognize phosphatidylserine on the cell surface. Here, we discuss the functions of Stabilin-1 and stabilin-2 as phosphatidylserine receptors in apoptotic cell clearance (efferocytosis) and cell fusion, and their ligand-recognition and signaling pathways.

## 1. Introduction

Phospholipids in the plasma membrane are asymmetrically maintained between the outer and inner leaflets of the lipid bilayer. Exposure of phosphatidylserine from the cytoplasmic leaflet to the exoplasmic leaflet at the plasma membrane by disruption of this phospholipid asymmetry is found in various biological systems and has diverse biological functions, including platelet activation [1,2], apoptosis [3,4], myoblast fusion [5,6], syncytiotrophoblast formation [7], immunoglobulin E-dependent stimulation of mast cells [8], lymphocyte activation [9], and regenerative axonal fusion [10,11].

Stabilin-1 (also referred to as Clever-1 or FEEL-1) and stabilin-2 (also referred to as HARE or FEEL-2) are type I membrane proteins containing an extracellular region with a similar domain composition [12]. Although each receptor has a different set of ligands that drive their respective biological processes, they share several biological functions, including endocytosis of acetylated low-density lipoprotein (LDL) [13,14] and advanced glycation end products (AGEs) [15], leukocyte trafficking [16,17,18], internalization of antisense oligonucleotides (ASOs) [19], and cell corpse clearance [20,21]. In particular, their function as phosphatidylserine-recognizing receptors is essential for apoptotic cell clearance, which in turn is necessary for maintaining tissue homeostasis and resolving inflammation. Inappropriate clearance of apoptotic cells contributes to the pathogenesis of various chronic inflammatory diseases, including autoimmune diseases [22], chronic obstructive pulmonary disease [23], and atherosclerosis [24]. In addition, a recent study showed that stabilin-2 contributes to myonuclear accretion during muscle regeneration, indicating its role in myoblast fusion [25]. In this review, we discuss the functions of stabilin proteins as phosphatidylserine receptors and address their phosphatidylserine-recognition domain and signaling pathways as seen as Table 1. In addition, we will discuss recent findings concerning the molecular mechanisms of phosphatidylserine externalization in apoptotic cells and myoblasts.

## 2. Apoptotic Cell Clearance

Billions of cells in diverse tissues throughout the body die daily through apoptosis, which is collectively responsible for homeostatic cell turnover in multicellular organisms. The rapid clearance of apoptotic cells is important for maintaining tissue homeostasis and preventing inappropriate inflammatory responses [26].

### 2.1. Phosphatidylserine Externalization in Apoptotic Cells

Cells undergoing apoptosis display “eat me” signals on the cell surface that serve to advertise their status and allow them to be efficiently removed by professional phagocytes (e.g., macrophages and immature dendritic cells) or non-professional specialized phagocytes (e.g., retinal pigment epithelial cells [27] and testicular Sertoli cells [28]). The most studied “eat me” signal is the externalization of phosphatidylserine from the cytoplasmic leaflet to the exoplasmic leaflet at the plasma membrane during apoptosis [29]. Phosphatidylserine externalization in the cell surface also has a role in apoptotic cell engulfment in *Caenorhabditis elegans* and *Drosophila* [30,31], indicating that phosphatidylserine externalization is among the evolutionarily conserved machineries for homeostatic removal of apoptotic cells. Scramblases mediate bidirectional translocation of phospholipids between the inner and outer leaflet of lipid bilayers. Two research groups recently demonstrated that ced-8 (cell death abnormality protein 8) and its mammalian ortholog Xkr8 (Xk-related protein 8) have scrambling activity that serves to expose phosphatidylserine on the apoptotic cell surface in a caspase-dependent manner [32,33], indicating that the scrambling mechanism of phosphatidylserine is conserved from lower organisms to mammals. Suzuki et al. showed that Xkr8 forms a functional complex with neuroplastin or basigin at the plasma membrane to execute scrambling activity in response to apoptotic stimuli [4]. Recently, Kawano et al. showed that Xkr8-null mice exhibit impaired apoptotic cell clearance and develop lupus-like autoimmune disease, as evidenced by increased levels of autoantibodies in serum and accumulation of immune complexes in glomeruli [34]. Conversely, lipid flippases transport aminophospholipids from the outer to the inner leaflet of the lipid bilayer to maintain membrane phospholipid asymmetry in healthy cells [35]. Segawa et al. showed that the P-type ATPase, ATP11C, which functions as a flippase to maintain membrane asymmetry, is inactivated by caspase-dependent cleavage in cells undergoing apoptosis, leading to phosphatidylserine externalization [36]. More recently, Sakuragi et al. showed that the loss of flippase activity increases the scrambling activity of Xkr8 [37]. Taken together, these observations indicate that surface exposure of phosphatidylserine on apoptotic cells is regulated by opposing scramblase and flippase activities during apoptosis and is an important signal for maintaining tissue homeostasis.

### 2.2. Apoptotic Cell Clearance by Stabilin Receptors

Phagocytes remove apoptotic cells by recognizing phosphatidylserine on the apoptotic cell surface; Phosphatidylserine receptors bind directly to phosphatidylserine of apoptotic cells, including TIM receptors (Tim-1 and Tim-4), Bai1, stabilin receptors, and CD300 receptors, whereas some receptors binds indirectly to apoptotic cells through soluble phosphatidylserine-binding proteins(e.g., Mfge8 and Gas6), including Tyro3/Axl/Mer (TAM) receptors (Tyro3, Axl, and Mer) and integrin receptors (αvβ3 and αvβ5) [38,39]. Stabilin-1 is mainly expressed in sinusoidal endothelial cells in spleen, lymph nodes, liver and adrenal cortex [40], and in alternatively activated macrophages [12,41,42] (also known as anti-inflammatory M2-like macrophages [43,44]). Park et al. demonstrated that stabilin-1 mediates apoptotic cell engulfment in alternatively activated macrophages in a phosphatidylserine-dependent manner [21]. Exogenous expression of stabilin-1 confers on mouse fibroblast L cells the ability to engulf damaged RBCs in a phosphatidylserine-dependent manner. In macrophages co-cultured with apoptotic cells, stabilin-1 is recruited to sites of recognition and engulfment of apoptotic cells. Blockade of stabilin-1 with an anti-stabilin-1 antibody or downregulation of stabilin-1 by short hairpin RNA (shRNA)-mediated knockdown has been shown to markedly inhibit engulfment of apoptotic cells by macrophages. In addition, extracellular acidic pH promotes the phagocytic activity of macrophages by increasing stabilin-1 expression [45]. Specifically, acidic pH increases expression of Ets-2 (E26 avian leukemia oncogene 2), which in turn binds to the stabilin-1 promoter to stimulate Stabilin-1 expression in macrophages, leading to increased phagocytic ability [45].

Stabilin-2 is expressed in sinusoidal endothelial cells in liver, spleen, lymph nodes and bone marrow [46,47], and in human monocyte-derived macrophages (HMDMs) [20]. Park et al. demonstrated that stabilin-2 is a membrane receptor that directly and stereospecifically recognizes phosphatidylserine on the surface of apoptotic cells during apoptotic cell engulfment [20]. Expression of stabilin-2 in mouse fibroblast L cells confers on transformed cells the ability to engulf apoptotic cells and phosphatidylserine-exposed red blood cells (RBCs) [20]. Masking of stabilin-2 using phosphatidylserine-containing liposomes causes marked inhibition of engulfment of phosphatidylserine-exposed RBCs and apoptotic cells by HMDMs or stabilin-2–expressing cells [20]. Intriguingly, engagement of HMDMs with an anti-stabilin-2 antibody was shown to cause production of the anti-inflammatory cytokine, transforming growth factor (TGF)-β, indicating that stabilin-2 mediates both tethering (phosphatidylserine recognition) and “tickling” (internalization of tethered corpses and activation of downstream signaling pathways) functions [20]. In addition, stabilin-2 was found to mediate phagocytosis of primary necrotic cells in a phosphatidylserine-dependent manner [48]. In addition to directly clearing apoptotic cells, Lee et al. demonstrated that stabilin-1 and stabilin-2 play a role in phosphatidylserine-dependent sequestration of damaged erythrocytes in hepatic sinusoidal endothelial cells (HSECs) [49]. They showed that damaged RBCs and phosphatidylserine-bearing beads were effectively sequestered in the hepatic sinusoid, an effect that was independent of Kupffer cells. Using a sinusoid-mimicking co-culture system, these authors showed that macrophages layered over HSECs efficiently engulfed damaged RBCs attached to HSECs, suggesting a cooperative role of both cell types in removing damaged RBCs in the hepatic sinusoid. Stabilin-1 and stabilin-2 in HSECs were shown to mediate tethering of damaged RBCs in a phosphatidylserine-dependent manner. Liver-specific knockdown of stabilin-1 and stabilin-2 in an in vivo animal model was shown to cause a marked reduction in the sequestration of phosphatidylserine-exposed RBCs in the hepatic sinusoid and a delay in their elimination from the circulation.

### 2.3. Phosphatidylserine-binding Domain in Stabilin Receptors

Stabilin-1 and stabilin-2 are composed of a large extracellular region, a transmembrane region, and a short cytoplasmic tail, see in Figure 1. The extracellular region of these receptors consists of four repeated clusters (I–IV). Clusters I–III consist of two atypical epidermal growth factor (EGF)-like domains, four EGF-like domains and two fasciclin I (FAS1) domains, whereas cluster IV is composed of two atypical EGF-like domains, three EGF-like domains, a FAS1 domain and a Link domain, as shown as in Figure 1 [50]. In stabilin-2, each cluster shares approximately 45% sequence homology and binds to phosphatidylserine through EGF-like domain repeats. The EGF-like domain repeats competitively impair apoptotic cell uptake by thioglycollate-elicited peritoneal macrophages in an in vivo animal model. At least four tandem repeats of the EGF-like domain containing an atypical EGF-like domain are required for binding to phosphatidylserine [50]. Moreover, calcium ions (Ca^2+^) are necessary for phosphatidylserine recognition of EGF-like domain repeats, such that a conserved asparagine^1407^ in the second atypical EGF-like domain serves as a critical residue for Ca^2+^ coordination and phosphatidylserine recognition, see in Figure 2 [51]. However, EGF-like domain repeats bind to phosphatidylserine with relatively low-affinity compared with Mfge8 (milk fat globule-EGF factor 8; also referred to as lactadherin) and calreticulin, which are high-affinity phosphatidylserine-binding proteins [52,53,54]. On the other hand, Kim et al. showed that the phosphatidylserine-binding activity of stabilin-2 is enhanced in acidic environments [51], suggesting that acidic pH might act as a danger signal that stimulates stabilin-2–mediated phagocytosis so as to resolve inflammation. On the basis of homology-based modeling of EGF-like domain repeats, the authors proposed the presence of a putative phosphatidylserine-binding loop (R^1405^CNQGPLGDGS^1415^) and found that a conserved histidine^1403^ in close proximity to the phosphatidylserine-binding loop plays a crucial role in enhancing the phagocytic capacity of stabilin-2 under acidic pH conditions, as seen as in Figure 2, suggesting that protonation of histidine^1403^ might induce a conformational change in the phosphatidylserine-binding loop that enhances binding affinity [51]. A detailed structural analysis of EGF-like domain repeats using X-ray crystallography will be needed to clarify the precise mechanism by which phosphatidylserine is recognized in a Ca^2+^-dependent or pH-dependent manner. Stabilin-1 also recognizes phosphatidylserine through its EGF-like domain repeats as stabilin-2 [21]. However, the precise mechanism responsible for phosphatidylserine binding remains to be determined.

### 2.4. Signaling Pathway Mediated by Stabilin Receptors

Once phagocytes recognize apoptotic targets, they initiate intracellular signals that cause cytoskeletal rearrangements, leading to internalization of apoptotic cells. Two representative signaling pathways that control apoptotic cell clearance were previously identified by genetic analyses in the nematode *C. elegans* [56]. The first pathway is composed of ced-1, ced-6, and ced-7, the mammalian homologs of which are MEGF10 (multiple EGF-like domains 10), Gulp1 (phosphotyrosine-binding domain-containing engulfment adaptor protein 1), and ABCA1 (ABC binding cassette subfamily A member 1), respectively [57,58,59]. The second pathway comprises ced-2, ced-5, ced-10 and ced-12, for which the corresponding mammalian homologs are CrkII (CRK adaptor protein), DOCK1 (dedicator of cytokinesis 1; also referred to as DOCK180), Rac1 (Rac family small GTPase 1), and ELMO1 (engulfment and cell motility 1), respectively [60,61,62]. Park et al. demonstrated that the Gulp1-dependent signaling pathway is required for stabilin-mediated phagocytosis [63,64]. Gulp1, composed of a phosphotyrosine-binding (PTB) domain, a leucine zipper domain and a proline-rich domain, acts as an adaptor protein to transduce a signal for cytoskeletal rearrangement in several phagocytic receptors, including MEGF10, LRP1 (LDL receptor-related protein 1), PEAR1 (platelet endothelial aggregation receptor 1; also referred to as MEGF12 and JEDI), and SR-BI (scavenger receptor class B member 1) [58,65,66,67]. Stabilin-1 and stabilin-2 bind directly to the PTB domain of Gulp1 through an NPxY/F motif in their cytoplasmic tail, see Figure 3 [63,64]. Knockdown of endogenous Gulp1 causes a substantial reduction in stabilin-1– and stabilin-2–mediated engulfment of phosphatidylserine-exposed RBCs. Conversely, overexpression of Gulp1 results in enhanced phagocytosis of phosphatidylserine-exposed RBCs by stabilin-2. The signaling pathway using ced-6 has been shown to converge on ced-10 in the nematode *C. elegans* [68]. Gulp1 was found to increase phosphorylation of MAPK p38 and activation of Rac1 in the signaling pathway from the scavenger receptor, SR-BI [67]. However, the signaling intermediates between Gulp1 and Rac1 in stabilin-mediated efferocytosis remain to be clarified. Intriguingly, stabilin-2 was also shown to indirectly activate the ELMO1-DOCK180-Rac1 pathway [69], a downstream signaling pathway engaged by several phagocytic receptors, including brain-specific angiogenesis inhibitor 1 (BAI1) and integrin αvβ5 receptor [70,71]. Stabilin-2 interacts with integrin β5 through its FAS1 domains, which are known to be involved in interactions between adhesion molecules and several integrin receptors [17,72,73], and transmits a Gulp1-independent signal through integrin αvβ5 that, in turn, recruits an ELMO1/DOCK180 complex, leading to Rac1 activation and subsequent phagocytosis. In Stabilin-2–expressing cells, knockdown of Gulp1 or DOCK180 partially inhibits phagocytic ability, whereas knockdown of both Gulp1 and DOCK180 causes additive inhibitory effects on phagocytosis [69], indicating that stabilin-2 acts through both Gulp1 and ELMO1/DOCK180 pathways, see in Figure 3. However, whether Stabilin-1 is linked to the ELMO1-DOCK180-Rac1 pathway remains to be investigated.

### 2.5. Regulation of Inflammation Following Efferocytosis

The efficient removal of apoptotic cells is important for maintenance of tissue homeostasis and protection of surrounding tissues from inappropriate inflammation [74]. It is known that interactions between macrophages and apoptotic cells actively suppress the production of inflammatory mediators [75]. Macrophages dampen inflammation and immune responses by stimulating secretion of anti-inflammatory and immunosuppressive cytokines, such as TGF-β [76], which is linked to the differentiation of immunosuppressive regulatory T cells [77]. The defective clearance of apoptotic cells by a deficiency of phagocytic receptors or phosphatidylserine-binding proteins was shown to cause lupus-like autoimmune diseases [22]. Park et al. showed that engagement of HMDMs with either phosphatidylserine-exposed RBCs or anti-stabilin-2 antibody causes secretion of TGF-β [20], a hallmark of the anti-inflammatory program associated with apoptotic cell recognition, indicating that stabilin-2 modulates inflammatory responses following efferocytosis. However, the molecular mechanism responsible for TGF-β secretion following Stabilin-2 activation will require further clarification. Additionally, whether stabilin-2–deficient mice exhibit defective apoptotic cell clearance and autoimmunity phenotype remains to be investigated. Several researchers have recently suggested that stabilin-1 has a role in regulating inflammatory and immune responses. Palani et al. demonstrated that stabilin-1 acts as an immunosuppressive molecule in monocytes, leading to suppression of lymphocyte activation [78]. Knockdown of stabilin-1 in monocytes causes increased production of inflammatory cytokines, such as tumor necrosis factor (TNF)-α, and allows T lymphocytes to produce more interferon (IFN)-γ and less interleukin (IL)-4 and IL-5 than control cells. Dunkel et al. showed that stabilin-1–null mice exhibit increased plasma levels of immunoglobulin G and M under resting conditions and augmented humoral immune responses after immunization in vivo [79]. Recently, Lee et al. demonstrated that stabilin-1 plays an important role in maintaining vascular integrity in sepsis [80], showing that stabilin-1 serves this protective function by promoting the clearance of apoptotic vascular endothelial cells damaged by severe inflammation. Consistent with this, stabilin-1 deficiency causes a decrease in survival in an animal model of sepsis, which is associated with reduced phagocytosis and increased vascular permeability. Furthermore, stabilin-1–mediated phagocytosis is inhibited by HMGB1 (high-mobility group box 1), a pro-inflammatory mediator of organ damage in sepsis [81]. Blockade of HMGB1 using a neutralizing antibody improves the phagocytic capacity of macrophages towards apoptotic cells. Collectively, these observations indicate that stabilin-1 and stabilin-2 play important roles in apoptotic cell clearance and subsequent resolution of inflammation. Therefore, therapeutic approaches designed to modulate the functions of stabilin receptors in diverse inflammatory diseases warrants further investigation.

## 3. Myoblast Fusion

Skeletal muscles comprise multinucleated myofibers formed by the fusion of mononucleated myoblasts; this fusion of myoblasts is required for embryonic myogenesis and skeletal muscle regeneration after muscle injury [82,83]. The following three-step model for facilitation of fusion pore formation was recently proposed: (1) recognition and adhesion of muscle cells, (2) achievement of close membrane proximity through the protrusive and resisting forces of the two fusion partners, and (3) destabilization of lipid bilayers [84,85,86]. Phosphatidylserine on the cell surface may be a candidate molecule involved in facilitating the destabilization of the lipid bilayer necessary for myoblast fusion. Furthermore, it is possible that the signal from phosphatidylserine-recognizing receptor regulates cytoskeletal rearrangement required for myoblast fusion.

### 3.1. Phosphatidylserine Externalization during Cell Fusion

It was previously reported that phosphatidylserine is transiently exposed at contact regions of myoblasts during differentiation in myoblast cultures [5] and at the cell surface of viable myoblasts in developing skeletal muscles [87]. Masking of phosphatidylserine with the phosphatidylserine-binding protein annexin V or an anti-phosphatidylserine antibody substantially abrogates myotube formation [5,88]. Moreover, cell surface phosphatidylserine has been implicated in other fusion models in mammals, including syncytiotrophoblast formation, osteoclast formation, and macrophage fusion. Phosphatidylserine is externalized during intercellular fusion of cytotrophoblasts to form syncytiotrophoblasts, a process that is inhibited by a monoclonal anti-phosphatidylserine antibody [89]. Recently, Verma et al. showed that fusion of osteoclast precursors depends on phosphatidylserine exposure on the surface of fusion-committed cells via a mechanism controlled by phosphatidylserine-regulated activities of several proteins, including DC-STAMP (dendrocyte expressed seven transmembrane protein) and annexins [90]. Cytokine-induced fusion of macrophages requires phosphatidylserine exposure on the cell surface and CD36-mediated lipid recognition [91]. Recently, some members of the TMEM16 protein family have been reported to act as Ca^2+^-dependent scramblases in non-apoptotic cells. Suzuki et al. demonstrated that TMEM16F (transmembrane protein F; also referred to as anoctamin 6) is dispensable for phosphatidylserine scrambling during apoptosis, but is required for the scramblase activity that externalizes phosphatidylserine on the surface in activated platelets [92,93]. Recent structural studies have provided insight into Ca^2+^-dependent activation and lipid transport by TMEM16 scramblase homologs [94,95,96]. Alvadia et al. suggested that two functions of TMEM16F as an ion channel and a scramblase are mediated by different protein conformation [95]. Feng et al. found that the PIP_2_-dependent conformational change of TMEM16F causes membrane distortion and thinning which facilitates phospholipid scrambling [96]. In studies of the phospholipid permeation pathway, Le et al. identified the inner activation gate of TMEM16F, showing that it is composed of three hydrophobic residues, phenylalanine^518^, tyrosine^563^, and isoleucine^612^ [97]. However, whether TMEM16F is responsible for phosphatidylserine exposure on the cell surface of myoblasts during fusion remains to be clarified. Recently, Whitlock et al. demonstrated that TMEM16E (also referred to as anoctamin 5) mediates Ca^2+^-dependent scrambling of phosphatidylserine in myogenic progenitor cells [6]. Myogenic progenitor cells from TMEM16E-deficient muscle exhibit defective myoblast fusion, which is associated with impaired phosphatidylserine exposure on the surface of TMEM16E-deficient myoblasts. Myofibers of TMEM16E-deficient mice also exhibit significantly fewer nuclei compared with those of wild-type mice. Thus, these observations indicate that TMEM16E is the likely candidate to mediate phosphatidylserine exposure on the cell surface of myoblasts. Structural studies of TMEM16E will be needed to clarify the precise mechanism by which phosphatidylserine is exposed on the cell surface of myoblasts. Furthermore, TMEM16E regulates osteoclast differentiation [98], and its gain-of-function mutation leads to gnathodiaphyseal dysplasia (GDD) with abnormal bone remodeling [99]. However, whether phospholipid scrambling activity of TMEM16E is involved in osteoclast fusion remains to be clarified.

### 3.2. Stabilin-2: A Phosphatidylserine Receptor in Myoblasts

Park et al. recently demonstrated that stabilin-2 acts as a phosphatidylserine receptor to contribute to myonuclear accretion during myogenic differentiation and muscle regeneration [25]. Forced expression of stabilin-2 led to increased myotube formation in the C2C12 myoblast cell line without affecting expression of myogenic factors. Furthermore, forced expression of stabilin-2 in mouse fibroblast L cells, which do not exhibit cell fusion in culture, allowed these cells to fuse with each other. Stabilin-2–deficient mice exhibit a reduction in skeletal muscle weight, myofiber size, and myonuclei number per individual myofiber. In a mouse model of regeneration, a stabilin-2 deficiency was shown to cause a delay in muscle repair after cardiotoxin-induced injury, as evidenced by smaller myofibers with fewer centralized nuclei in regenerating muscles. A stabilin-2 deficiency does not affect proliferation or activation of myogenic progenitor cells (also known as satellite cells) during muscle regeneration, indicating a role for stabilin-2 in myoblast fusion rather than myogenic differentiation. In agreement with the in vivo phenotype, stabilin-2–null myoblasts exhibit a reduction in myoblast fusion compared with wild-type myoblasts and a loss of phosphatidylserine dependence. However, the fusion defect in stabilin-2–deficient mice is mild compared with that observed following ablation of Myomaker or Myomixer (also referred to as Myomerger or Minion), which were recently identified as muscle-specific proteins necessary for myoblast fusion [100,101,102]. This relatively mild phenotype might be explained by the redundancy of phosphatidylserine receptors and their signaling pathways, as shown in efferocytosis.

Park et al. also showed that stabilin-2 expression was increased in myoblasts during differentiation through calcineurin signaling [25]. During myogenic differentiation, activated calcineurin dephosphorylates NFATc1 (nuclear factor of activated T cells c1) [103], which subsequently migrates into the nucleus and binds to the proximal region of the stabilin-2 promoter, leading to increased stabilin-2 expression. Several recent studies have provided support for the role of calcineurin-NFATc1 signaling in stabilin-2 expression in skeletal muscles. Ectopic expression of sarcolipin (Sln), a regulator of sarcoplasmic reticulum Ca^2+^-ATPase, which activates calcineurin-NFATc1 signaling, causes an increase in strabilin-2 protein content in functionally overloaded plantaris muscles [104]. Stabilin-2 expression was found to be reduced in mdx/Sln-knockout muscles, in which the calcineurin signaling pathway is impaired [105].

Actin cytoskeleton regulator proteins are important for myoblast fusion as well as efferocytosis, as evidenced by the fact that phagocytic pathway components, such as ELMO, CrkII, DOCK1, and Rac1, are required for myoblast fusion [106,107,108]. This raises the possibility that the interaction between stabilin-2 and phosphatidylserine creates a signal that recruits fusion-mediating molecules, leading to the cytoskeletal rearrangements necessary for myoblast fusion. However, the molecular details of the mechanism by which stabilin-2 mediates phosphatidylserine-dependent fusion remain to be clarified. Two G protein-coupled receptor (GPCR) proteins, BAI1 and BAI3, were recently identified as phosphatidylserine-recognition receptors that promote myoblast fusion through the ELMO-DOCK1-Rac1 pathway [106,109]. Hamoud et al. demonstrated that stabilin-2 is functionally associated with BAI3 during myoblast fusion. The authors showed that stabilin-2 promotes the GPCR activity of BAI3 and subsequent activation of heterotrimeric G-proteins, which then contribute to the recruitment of BAI3-binding ELMO proteins, leading to stabilization of BAI3 [110].

## 4. Conclusions and Perspectives

Over the past two decades, enormous strides have been made in unveiling the phosphatidylserine receptors that recognize phosphatidylserine in various biological systems, leading to a remarkable increase in our understanding of efferocytosis. Intriguingly, a role for phosphatidylserine receptors in myoblast fusion has recently emerged. However, unanswered questions concerning their action mechanism remain. One of the most pressing questions is how phosphatidylserine receptors such as stabilin-2 mediate the different cellular responses, efferocytosis and cell fusion, through phosphatidylserine recognition. For example, it has been shown that engagement of macrophages with phosphatidylserine-containing liposomes inhibits apoptotic cell engulfment [20,111], whereas phosphatidylserine-containing liposomes promote cell fusion in myoblasts [25,88]. Stabilin-1 and stabilin-2 also act as scavenger receptors with multiple functions: Stabilin-1 acts as a receptor for SPARC (secreted protein acidic and rich in cysteine) and acetylated LDL [14,112], whereas stabilin-2 acts as a receptor for hyaluronic acids, chondroitin sulfate, and heparin [113,114,115]. Other intriguing questions include: how do stabilin-1 and stabilin-2 accomplish multiple tasks, and what are the regulatory mechanisms and signaling pathways for different ligands? Answering these questions requires a better understanding of the signaling intermediates downstream of phosphatidylserine receptors in various biological processes and the regulatory mechanisms that modify their signaling pathways. Furthermore, the functions of phosphatidylserine receptors need to be elucidated in various biological processes, such as viral infection [116], immune responses [117], and axon regeneration [10]. A better understanding of stabilin receptors and their action mechanism could be useful for the development of therapeutic strategies for controlling diseases associated with cell surface exposure of phosphatidylserine.

## Figures and Tables

**Figure 1 biomolecules-09-00387-f001:**
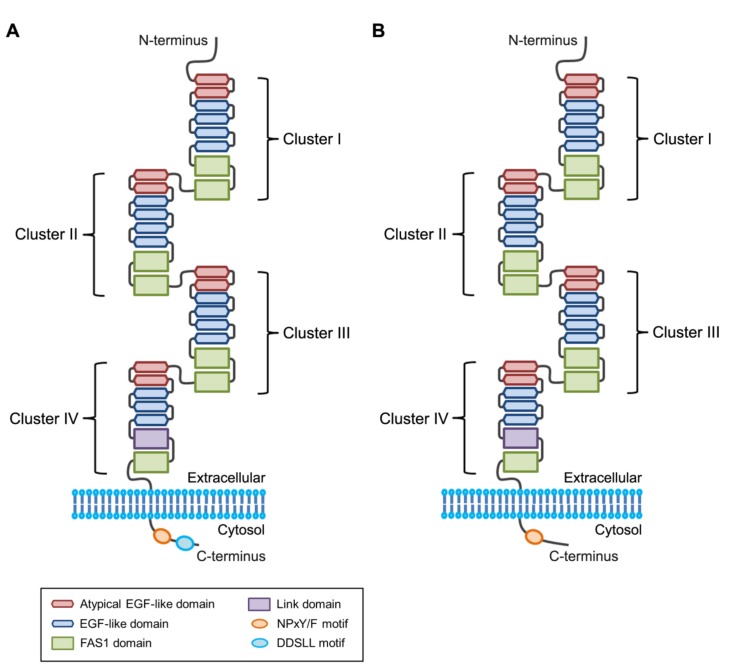
Structure of stabilin receptors. The extracellular regions of stabilin-1 (**A**) and stabilin-2 (**B**) are composed of four clusters containing epidermal growth factor (EGF)-like domain repeats, which bind to phosphatidylserine on the apoptotic cell surface. The crystal structure of fasciclin I (FAS1) domain of stabilin-2 has recently been determined [55]. The cytosolic region of stabilin-1 contains an asparagine-proline-x-phenylalanine (NPxF) motif (where x is any amino acid) and a aspartate-aspartate-serine-leucine-leucine (DDSLL) motif, whereas the cytosolic region of stabilin-2 contains an asparagine-proline-x-tyrosine (NPxY) motif (where x is any amino acid).

**Figure 2 biomolecules-09-00387-f002:**
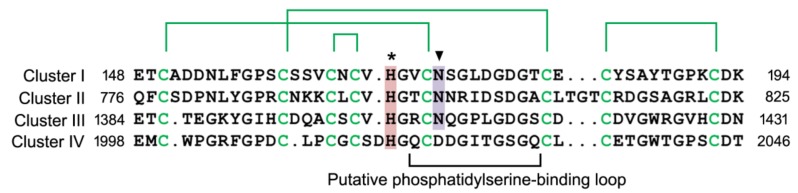
The amino acid sequences of the second atypical EGF-like domains in the four clusters of stabilin-2. Disulfide bonds in atypical EGF-like domain are indicated by green lines. Arrowhead indicates conserved asparagine, which is important for phosphatidylserine recognition by a coordinating Ca^2+^ ion. Asterisk indicates conserved histidine that is critical for enhancement of phagocytic activity in low pH environments.

**Figure 3 biomolecules-09-00387-f003:**
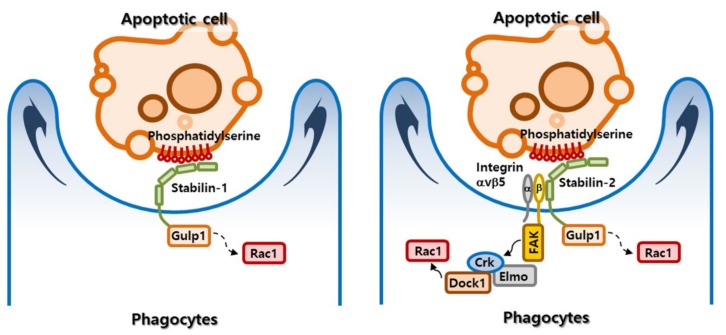
Signaling pathways elicited by stabilin-mediated phagocytosis of apoptotic cells. Stabilin-1 and stabilin-2 directly binds to phosphatidylserine on the surface of apoptotic cells, resulting in activation of Rac1 (Rac family small GTPase 1), which reorganizes the actin cytoskeleton for apoptotic cell engulfment via a Gulp1-dependent mechanism. However, intermediate signaling molecules between Gulp1 (phosphotyrosine-binding domain-containing engulfment adaptor protein 1) and Rac1 remain to be characterized. In the Gulp1-independent pathway, stabilin-2 interacts through its fasciclin I (FAS1) domains with integrin β5, which in turn activates focal adhesion kinase (FAK) and recruits the CrkII (CRK adaptor protein)-ELMO1 (engulfment and cell motility 1)-DOCK1 (dedicator of cytokinesis 1; also referred to as DOCK180) complex, leading to activation of Rac1 and induction of cytoskeletal rearrangement.

**Table 1 biomolecules-09-00387-t001:** The roles of stabilin-1 and stabilin-2 as phosphatidylserine receptors.

Receptor	Details and Comments	Ref.
Stabilin-1	Stabilin-1 mediates phagocytosis of aged red blood cells (RBCs) and apoptotic cells in alternatively activated macrophages in a phosphatidylserine-dependent manner. Stabilin-1 interacts with phosphatidylserine through its epidermal growth factor (EGF)-like domain repeats.	[21]
	Gulp1 (phosphotyrosine-binding domain-containing engulfment adaptor protein 1) functions downstream of stabilin-1 receptor to remove aged RBCs. Stabilin-1 binds to phosphotyrosine-binding domain of Gulp1 via its asparagine-proline-x-phenylalanine (NPxF) motif (where x is any amino acid).	[64]
	Stabilin-1 enhances phosphatidylserine-dependent erythrophagocytosis through hepatic sequestration of damaged RBC. Knockdown of stabilin-1 and stabilin-2 delays hepatic clearance of damaged RBCs in vivo.	[49]
	Acidic pH enhances phagocytosis of aged RBCs through enhancing stabilin-1 expression. Ets-2 (E26 avian leukemia oncogene 2) acts as a positive regulator to regulate stabilin-1 expression.	[45]
	Stabilin-1-mediated phagocytosis plays an important role in maintaining vascular integrity during sepsis. Stabilin-1 deficiency promotes disease progression caused by septic shock.	[80]
Stabilin-2	Stabilin-2 mediates phagocytosis of aged RBCs and apoptotic cells in human monocyte-derived macrophages in a phosphatidylserine-dependent manner. Stabilin-2 activation stimulates transforming growth factor (TGF)-β production.	[20]
	Gulp1 functions downstream of stabilin-2 receptor to effectively clear aged RBCs. Stabilin-2 binds to phosphotyrosine-binding domain of Gulp1 via its asparagine-proline-x- tyrosine (NPxY) motif (where x is any amino acid).	[63]
	Stabilin-2 interacts with phosphatidylserine through its EGF-like domain repeats. Atypical EGF-like domains in Stabilin-2 play an important role in phosphatidylserine binding.	[50]
	Extracellular acidic pH enhances stabilin-2-mediated efferocytosis. The conserved histidine in atypical EGF-like domain modulates the phosphatidylserine-binding affinity of stabilin-2.	[51]
	Stabilin-2 enhances phosphatidylserine-dependent erythrophagocytosis through hepatic sequestration of damaged RBCs. Knockdown of stabilin-1 and stabilin-2 delays hepatic clearance of damaged RBCs in vivo.	[49]
	Stabilin-2 binds to integrin β5 through its fasciclin I (FAS1) domains. Integrin αvβ5 and its signaling pathway are involved in stabilin-2–mediated phagocytosis.	[69]
	Stabilin-2 modulates efficiency of myoblast fusion during myogenic differentiation. Stabilin-2–deficient mice display impaired muscle regeneration.	[25]

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
