# Peer review of "Stabilin Receptors: Role as Phosphatidylserine Receptors"

_biomolecules, 2019, doi:10.3390/biom9080387_

Round 1

Reviewer 1 Report

This review by Park and Kim discusses the recent advances in understanding of the roles of Stabilin-1 and -2 receptors for the negatively charged lipid phosphatidylserine (PS). PS externalization is a key signal in cellular pathways controlling processes that range from apoptosis and cell clearance to platelet activation and cell fusion.

The review is well written, clear and informative. I have only relatively minor suggestions to improve clarity in some points:

Are the pathways controlled by Stabilin-1 activation know and characterized? If so, the authors could add a panel to Fig. 3 describing them. If not, a clear statement to the fact that further work is needed would be helpful to a reader to understand where the open questions in the field lie. 2, Line 73, the authors state that “More recently, Sakuragi et al. showed that the scrambling activity of Xkr8 is also regulated by flippase activity”. My understanding is that the work by Sakuragi et al shows that the two are competing and compensating each other and that phosphorilation regulates xkr8 activity and that of the flippase. 7, Line 263, The authors should also briefly discuss the recent structural work on TMEM16F by Alvadia et al., Elife 2019 and by Feng et al., Cell Reports 2019. 6-7, paragraph 3.1 This paragraph should have a summarizing concluding sentence. As it stands a reader is left hanging with what is the take-home message of this section. The evidence reported by the authors suggest that TMEM16E, rather than TMEM16F, is the likely candidate to mediate PS exposure in myoblasts. Indeed, TMEM16F KO animals do not display muscle phenotypes. This should be discussed. The authors mention that PS exposure is important in bone formation. It would be interesting to briefly discuss that gain of function mutations of TMEM16E also lead to a bone malformation disorder, GDD (see Di Zanni et al., Cell Mol Life Sci, 2018).

Minor:

In several places, the authors write “exonal” instead of “axonal” or “MEM16E” instead of “TMEM16E”. Please correct.

Pg.2, Line 53 The authors state: “The most likely “eat me” signal is…” what does most likely mean in this context? Most common?

Pg. 3, Line 140: The authors state: “A putative phosphatidylserine loop…” do they mean “A putative phosphatidylserine binding loop…”?

Pg. 8, Line 335: The authors state: “…controlling diseases associated with cell surface phosphatidylserine.” Do they mean “…controlling diseases associated with cell surface exposure of phosphatidylserine.”

Author Response

Comments and Suggestions for Authors (Reviewer 1)

This review by Park and Kim discusses the recent advances in understanding of the roles of Stabilin-1 and -2 receptors for the negatively charged lipid phosphatidylserine (PS). PS externalization is a key signal in cellular pathways controlling processes that range from apoptosis and cell clearance to platelet activation and cell fusion.

The review is well written, clear and informative. I have only relatively minor suggestions to improve clarity in some points:

Are the pathways controlled by Stabilin-1 activation know and characterized? If so, the authors could add a panel to Fig. 3 describing them. If not, a clear statement to the fact that further work is needed would be helpful to a reader to understand where the open questions in the field lie.

[Answer]

Signaling pathway by Stabilin-1-mediated phagocytosis was added in Figure 3 (left panel).

2, Line 73, the authors state that “More recently, Sakuragi et al. showed that the scrambling activity of Xkr8 is also regulated by flippase activity”. My understanding is that the work by Sakuragi et al shows that the two are competing and compensating each other and that phosphorilation regulates xkr8 activity and that of the flippase.

[Answer]

In line 77, “the scrambling activity of Xkr8 is also regulated by flippase activity” was changed into “loss of flippase activity increases the scrambling activity of Xkr8.”

7, Line 263, The authors should also briefly discuss the recent structural work on TMEM16F by Alvadia et al., Elife 2019 and by Feng et al., Cell Reports 2019.

[Answer]

The brief discussion for recent studies by Alvaldia et al. and Feng et al. was added in lines 296-299.

6-7, paragraph 3.1 This paragraph should have a summarizing concluding sentence. As it stands a reader is left hanging with what is the take-home message of this section. The evidence reported by the authors suggest that TMEM16E, rather than TMEM16F, is the likely candidate to mediate PS exposure in myoblasts. Indeed, TMEM16F KO animals do not display muscle phenotypes. This should be discussed.

[Answer]

In paragraph 3.1, a summarizing concluding sentence was added in lines 309-312.

The authors mention that PS exposure is important in bone formation. It would be interesting to briefly discuss that gain of function mutations of TMEM16E also lead to a bone malformation disorder, GDD (see Di Zanni et al., Cell Mol Life Sci, 2018).

[Answer]

The brief discussion about gain of function mutation of TMEM16E was added in lines 312-315.

Minor:

In several places, the authors write “exonal” instead of “axonal” or “MEM16E” instead of “TMEM16E”. Please correct.

[Answer]

In lines 15 and 28, “exonal” was changed into “axonal.” In line 376, “exon” was changed into “axon.” In lines 306 and 307, “MEM16E” was changed into “TMEM16E.”

Pg.2, Line 53 The authors state: “The most likely “eat me” signal is…” what does most likely mean in this context? Most common?

[Answer]

In line 55, “most likely” was changed into “most studied.”

Pg. 3, Line 140: The authors state: “A putative phosphatidylserine loop…” do they mean “A putative phosphatidylserine binding loop…”?

[Answer]

In line 164, “A putative phosphatidylserine loop” was changed into “A putative phosphatidylserine binding loop.”

Pg. 8, Line 335: The authors state: “…controlling diseases associated with cell surface phosphatidylserine.” Do they mean “…controlling diseases associated with cell surface exposure of phosphatidylserine.”

[Answer]

In line 378, “cell surface phosphatidylserine” was changed into “cell surface exposure of phosphatidylserine.”

Reviewer 2 Report

The manuscript by Seung-Yoon Park and In-San Kim entitled “Stabilin receptors: Role as phosphatidylserine receptors” concentrates on the stabilin receptors (stabilin-1 and stabilin-2) in the context of the phosphatidylserine recognition and clearance of phosphatidylserine-exposing cells. The paper presents a brief introduction, the stabilin involvement in apoptotic cell engulfment and roles in myoblast fusion.

In general, the manuscript is comprehensive, concisely written and helpful, especially that a review about stabilin receptors was missing so far in the literature. I believe it will be attractive to the readership of the Biomolecules. I therefore recommend the acceptance of the manuscript and suggest a few revisions:

- I feel that slightly more information about the ligands internalized by stabilin-2 (e.g. chondroitin sulfates, advanced glycation end products, oligonucleotides) would underline its complex role in physiological processes;

 - I think it would be beneficial to briefly inform the reader about other receptors that are responsible for the phosphatidylserine recognition (e.g. TAM receptors, TIM1/4 receptors);

- the authors might mention in Figure 1 that the crystal structure of the FAS1 domain of stabilin-2 has recently been determined (Twarda-Clapa et al., (2018). Crystal structure of the FAS1 domain of the hyaluronic acid receptor stabilin‐2. Acta Cryst. D74, 695–701; https://doi.org/10.1107/S2059798318007271).

- The authors are encouraged to check the manuscript for typos (line 210: “shown to causes”)

Author Response

Comments and Suggestions for Authors (Reviewer 2)

The manuscript by Seung-Yoon Park and In-San Kim entitled “Stabilin receptors: Role as phosphatidylserine receptors” concentrates on the stabilin receptors (stabilin-1 and stabilin-2) in the context of the phosphatidylserine recognition and clearance of phosphatidylserine-exposing cells. The paper presents a brief introduction, the stabilin involvement in apoptotic cell engulfment and roles in myoblast fusion.

 In general, the manuscript is comprehensive, concisely written and helpful, especially that a review about stabilin receptors was missing so far in the literature. I believe it will be attractive to the readership of the Biomolecules. I therefore recommend the acceptance of the manuscript and suggest a few revisions:

- I feel that slightly more information about the ligands internalized by stabilin-2 (e.g. chondroitin sulfates, advanced glycation end products, oligonucleotides) would underline its complex role in physiological processes;

[Answer]

The information about ligands internalized by Stabilin-2 was added in lines 32-35 and 369-370.

 - I think it would be beneficial to briefly inform the reader about other receptors that are responsible for the phosphatidylserine recognition (e.g. TAM receptors, TIM1/4 receptors);

[Answer]

The brief information about other receptors that are responsible for the phosphatidylserine recognition was added in lines 83-87.

- the authors might mention in Figure 1 that the crystal structure of the FAS1 domain of stabilin-2 has recently been determined (Twarda-Clapa et al., (2018). Crystal structure of the FAS1 domain of the hyaluronic acid receptor stabilin‐2. Acta Cryst. D74, 695–701; https://doi.org/10.1107/S2059798318007271).

[Answer]

“The crystal structure of the FAS1 domain of stabilin-2 has recently been determined” was inserted into the legend of Figure 1 (lines 177-178).

- The authors are encouraged to check the manuscript for typos (line 210: “shown to causes”)

[Answer]

In line 240, “shown to causes” was changed into “shown to cause.”

Reviewer 3 Report

This review by Park and Kim nicely summarized the functions of phosphatidylserine receptors stabilin-1 and stabilin-2 in efferocytosis and cell fusion. It properly accounts the current representative research in this field. There are several issues that need to be refined and/or clarified.

line 34: “In particular, their function as phosphatidylserine-recognizing receptors is essential for apoptotic cell clearance, which in turn is necessary for maintaining tissue homeostasis and resolving inflammation. This function is also necessary for cell fusion, an important event in the development of tissues and organs such as skeletal muscles.”

Please add references here. Phosphatidylserine receptors play important role in myoblast fusion. However, to my knowledge, there is no direct evidence showed that “phosphatidylserine-recognizing receptors” is necessary for cell fusion currently. Is it possible that phosphatidylserine externalization triggers  other cell signaling instead of recognizing PS receptors which play an important role in cell fusion? 

Line 50-76. It’ll be nice to briefly introduce the readers the definitions of lipid flippases and scramblases and their differential functions on transporting PS.

Line 85-88. Please clarify the reference(s) for these two sentences.

Line 108: grammar “...which are participate”

line 269, 270 misspelled TMEM16E “MEM16E-deficient muscle exhibit defective myoblast fusion, which is associated with impaired phosphatidylserine exposure on the surface of MEM16E-deficient myoblasts. Myofibers of TMEM16E-deficient mice also exhibit significantly fewer nuclei compared with those of wild-type mice.”

6. Since efferocytosis and cell fusion are different biological process, adding a discussion on the differential roles of phosphatidylserine receptors in efferocytosis and cell fusion will make the manuscript more attractive to the readers.

Author Response

Comments and Suggestions for Authors (Reviewer 3)

This review by Park and Kim nicely summarized the functions of phosphatidylserine receptors stabilin-1 and stabilin-2 in efferocytosis and cell fusion. It properly accounts the current representative research in this field. There are several issues that need to be refined and/or clarified.

line 34: “In particular, their function as phosphatidylserine-recognizing receptors is essential for apoptotic cell clearance, which in turn is necessary for maintaining tissue homeostasis and resolving inflammation. This function is also necessary for cell fusion, an important event in the development of tissues and organs such as skeletal muscles.” Please add references here.

[Answer]

“This function is also necessary for cell fusion, an important event in the development of tissues and organs such as skeletal muscles” was deleted from the text. The role of Stabilin-2 in myoblast fusion was briefly described, and reference was added (lines 41-42).

Phosphatidylserine receptors play important role in myoblast fusion. However, to my knowledge, there is no direct evidence showed that “phosphatidylserine-recognizing receptors” is necessary for cell fusion currently. Is it possible that phosphatidylserine externalization triggers other cell signaling instead of recognizing PS receptors which play an important role in cell fusion? 

[Answer]

It has recently reported that several phosphatidylserine receptors, such as Bai1, Bai3, and Stabilin-2, are involved in myoblast fusion. Although their precise mechanism remains to be clarified, the signal from phosphatidylserine receptors may trigger cytoskeletal rearrangement required for myoblast fusion. Besides to the signal through phosphatidylserine receptors, it is possible that cell surface phosphatidylserine triggers other cell signaling for cell fusion.

 Line 50-76. It’ll be nice to briefly introduce the readers the definitions of lipid flippases and scramblases and their differential functions on transporting PS.

[Answer]

The definition of scramblase was added in lines 60-61. The definition of lipid flippase was described in lines 72-74.

Line 85-88. Please clarify the reference(s) for these two sentences.

[Answer]

The reference for two sentences was added (lines 110 and 112).

Line 108: grammar “...which are participate”

[Answer]

In line 91, “Which are participate” was changed into “which are participated.”

line269, 270 misspelled TMEM16E “MEM16E-deficient muscle exhibit defective myoblast fusion, which is associated with impaired phosphatidylserine exposure on the surface of MEM16E-deficient myoblasts. Myofibers of TMEM16E-deficient mice also exhibit significantly fewer nuclei compared with those of wild-type mice.”

[Answer]

In lines 306 and 307, “MEM16E” was changed into “TMEM16E.”

Since efferocytosis and cell fusion are different biological process, adding a discussion on the differential roles of phosphatidylserine receptors in efferocytosis and cell fusion will make the manuscript more attractive to the readers.

[Answer]

The role of phosphatidylserine receptors in efferocytosis is described in lines 83-89. The possible role of phosphatidylserine receptors in myoblast fusion was added in lines 274-275.

Reviewer 4 Report

Major comments:

1. The manuscript describes the role of stabilins as phosphatidylserine receptors. Currently, there are two stabilin receptors reported, stabilin-1 and stabilin-2. Overall most of the text was about stabilin-2. I admit the fact that there are not so much studies of stabilin-1 as a phosphatidylserine receptor compared to stabilin-2. However, the title states both molecules, so there should not be so huge unbalance in those parts where the literature is available. For example paragraph “2-3. Phosphatidylserine-binding domain in stabilin receptors” merely describes stabilin-2. and Figure 1. illustrates the structure of stabilin-2. In the figure there could be both stabilins side by side and in the text more of stabilin-1 (in this particular paragraph: stabilin-1 also binds to phosphatidylserine through EGF-like domain repeats as stabilin-2, ref 20 in the article by the writers of this manuscript).

2. Lines 297-300: “Ectopic expression of sarcolipin (Sln), a regulator of sarcoplasmic reticulum Ca2+-ATPase, which activates calcineurin-NFATc1 signaling, causes an increase in strabilin-2 protein content in functionally overloaded plantaris muscles [99].” I did not find such a statement in the original article, either of stabilin-2 or strabilin-2. What molecule the writers mean or is the reference incorrect?

3. A chart / table that summarizes the role of stabilins as phosphatidylserine receptors would be nice illustrative addition to the text.

Minor comments:

1. I would prefer that in the text the description of stabilin receptors would be in numerical order (for example in the paragraph 2.2.).

2. Rephrase the text in the line 108 “Which are participate” and in the line 241 “is likely one”.

3. In the line 435 “33. omorski, T.” The first letter, P, is missing.

Author Response

Comments and Suggestions for Authors (Reviewer 4)

Major comments:

The manuscript describes the role of stabilins as phosphatidylserine receptors. Currently, there are two stabilin receptors reported, stabilin-1 and stabilin-2. Overall most of the text was about stabilin-2. I admit the fact that there are not so much studies of stabilin-1 as a phosphatidylserine receptor compared to stabilin-2. However, the title states both molecules, so there should not be so huge unbalance in those parts where the literature is available. For example paragraph “2-3. Phosphatidylserine-binding domain in stabilin receptors” merely describes stabilin-2. and Figure 1. illustrates the structure of stabilin-2. In the figure there could be both stabilins side by side and in the text more of stabilin-1 (in this particular paragraph: stabilin-1 also binds to phosphatidylserine through EGF-like domain repeats as stabilin-2, ref 20 in the article by the writers of this manuscript).

[Answer]

As the reviewer has suggested, the structure of stabilin-1 was added in Figure 1 (left panel). The information about the phosphatidylserine-binding domain of stabilin-1 was described in the text (lines 171-173).

Lines 297-300: “Ectopic expression of sarcolipin (Sln), a regulator of sarcoplasmic reticulum Ca2+-ATPase, which activates calcineurin-NFATc1 signaling, causes an increase in strabilin-2 protein content in functionally overloaded plantaris muscles [99].” I did not find such a statement in the original article, either of stabilin-2 or strabilin-2. What molecule the writers mean or is the reference incorrect?

[Answer]

This reference is incorrect. In line 343, ref.99 in original manuscript was replaced with a correct reference (ref. 104; Fajardo, V.A.; Rietze, B.A.; Chambers, P.J.; Bellissimo, C.; Bombardier, E.; Quadrilatero, J.; Tupling, A.R. Effects of sarcolipin deletion on skeletal muscle adaptive responses to functional overload and unload. Am J Physiol Cell Physiol 2017, 313, C154-C161, doi:10.1152/ajpcell.00291.2016).

A chart / table that summarizes the role of stabilins as phosphatidylserine receptors would be nice illustrative addition to the text.

[Answer]

The roles of Stabilin-1 and Stabilin-2 as phosphatidylserine receptors were summarized in Table 1 (page 17).

Minor comments:

I would prefer that in the text the description of stabilin receptors would be in numerical order (for example in the paragraph 2.2.).

[Answer]

In section 2.2., the description for Stabilin-1 in lines 130-143 was moved into lines 88-102.

The sentence in lines 106-108 was moved into lines 103-104.

Rephrase the text in the line 108 “Which are participate” and in the line 241 “is likely one”.

[Answer]

In line 91, “which are participate” was changed into “which are participated.”

In line 272, “is likely one candidate” was changed into “may be a candidate.”

In the line 435 “33. omorski, T.” The first letter, P, is missing.

[Answer]

In line 489 (ref. 35), “omorski, T” was changed into “Pomorski, T.”